# Historical Evolutionary Dynamics and Phylogeography Analysis of Transmissible Gastroenteritis Virus and Porcine Deltacoronavirus: Findings from 59 Suspected Swine Viral Samples from China

**DOI:** 10.3390/ijms23179786

**Published:** 2022-08-29

**Authors:** Quanhui Yan, Keke Wu, Weijun Zeng, Shu Yu, Yuwan Li, Yawei Sun, Xiaodi Liu, Yang Ruan, Juncong Huang, Hongxing Ding, Lin Yi, Mingqiu Zhao, Jinding Chen, Shuangqi Fan

**Affiliations:** 1College of Veterinary Medicine, South China Agricultural University, Guangzhou 510642, China; 2Guangdong Laboratory for Lingnan Modern Agriculture, Guangzhou 510642, China; 3Key Laboratory of Zoonosis Prevention and Control of Guangdong Province, Guangzhou 510642, China

**Keywords:** coronavirus, TGEV, PDCoV, evolutionary dynamics, phylogeography, Bayesian inference

## Abstract

Since the beginning of the 21st century, humans have experienced three coronavirus pandemics, all of which were transmitted to humans via animals. Recent studies have found that porcine deltacoronavirus (PDCoV) can infect humans, so swine enteric coronavirus (SeCoV) may cause harm through cross-species transmission. Transmissible gastroenteritis virus (TGEV) and PDCoV have caused tremendous damage and loss to the pig industry around the world. Therefore, we analyzed the genome sequence data of these two SeCoVs by evolutionary dynamics and phylogeography, revealing the genetic diversity and spatiotemporal distribution characteristics. Maximum likelihood and Bayesian inference analysis showed that TGEV could be divided into two different genotypes, and PDCoV could be divided into four main lineages. Based on the analysis results inferred by phylogeography, we inferred that TGEV might originate from America, PDCoV might originate from Asia, and different migration events had different migration rates. In addition, we also identified positive selection sites of spike protein in TGEV and PDCoV, indicating that the above sites play an essential role in promoting membrane fusion to achieve adaptive evolution. In a word, TGEV and PDCoV are the past and future of SeCoV, and the relatively smooth transmission rate of TGEV and the increasing transmission events of PDCoV are their respective transmission characteristics. Our results provide new insights into the evolutionary characteristics and transmission diversity of these SeCoVs, highlighting the potential for cross-species transmission of SeCoV and the importance of enhanced surveillance and biosecurity measures for SeCoV in the context of the COVID-19 epidemic.

## 1. Introduction

Coronaviruses (CoVs) are enveloped viruses with a positive-sense, single-stranded RNA genome. They are members of the Nidovirales order, family Coronaviridae, and subfamily Coronaviridae, as defined by the International Committee on Taxonomy of Viruses [1]. With a genome length of 27–32 kilobases, CoVs are the largest of all known RNA viruses. The Coronavirinae can be divided into four genera, alpha-, beta-, gamma- and deltacoronaviruses, based on serological and genomic approaches [2]. Some betacoronaviruses can cause respiratory syndromes in hosts, such as severe acute respiratory syndrome coronavirus (SARS-CoV), Middle East respiratory syndrome coronavirus (MERS-CoV), and severe acute respiratory syndrome coronavirus 2 (SARS-CoV-2), which has become a global pandemic in recent years [3,4]. Coronavirus such as MERS-CoV, the spill over into humans via camels due to the cross-species transmission mechanisms [5], caused epidemiological investigation and traceability of coronaviruses to be a priority. Notably, a recent study reported the world’s first case of cat-to-human transmission of SARS-CoV-2 [6]. Previously, two captive animals, mink and hamster, as well as wild white-tailed deer in North America were capable of transmitting SARS-CoV-2 to humans [7,8].

Swine enteric coronaviruses (SeCoVs) are coronaviruses with swine intestinal tissue tropism, causing watery diarrhea, vomiting, and even death in sows and piglets [9]. At present, four SeCoVs have been identified, namely transmissible gastroenteritis virus (TGEV), porcine epidemic diarrhea virus (PEDV), porcine deltacoronavirus (PDCoV), and severe acute diarrhea syndrome coronavirus (SADS-CoV). Besides PDCoV, all other SeCoVs are alphacoronaviruses [10].

TGEV has been spreading in swine herds for decades, since its first outbreak in the 1940s in the U.S. TGEV has also been reported in other countries, including France, Germany, China, Japan, South Africa, and Brazil, causing significant damage to the pig industry worldwide [11]. In addition, the N-terminal domain (NTD) of the TGEV spike (S) protein presents as a 621–681 nt deletion in some cases, resulting in intestinal tissue tropism and respiratory tissue tropism, and such strains are considered to be variants of TGEV, called porcine respiratory coronaviruses (PRCV) [12,13]. However, in the past few years, coronaviruses have emerged from chimeric TGEV and PEDV in some European countries [14], suggesting that TGEV chimeric viruses may also occur in other countries.

As a newly emerged SeCoV, PDCoV was first identified in Hong Kong in 2012 [15]. Due to the clinical signs caused by PDCoV being similar to TGEV, the risk of PDCoV to the pig industry was not initially noticed. A survey of diarrhea samples in Ohio in 2014 found that the PDCoV detection rate was more than 90% [16], followed by significant outbreaks of PDCoV in the United States [17], Canada [18], South Korea [19], and other countries.

Lau et al. demonstrated that PDCoV HKU15 (GenBank: JQ065042) is closely related to the genome of sparrow coronavirus HKU17 (GenBank: NC_016992), which is also a deltacoronavirus, and that HKU15 was the product of recombination between HKU17 and bulbul coronavirus HKU11 (GenBank: FJ376619) in 2018 [20]. Since the GC content of the PDCoV genome is slightly lower than that of sparrow coronavirus (Sp-CoV), this may facilitate the adaptability of avian-derived viruses to replicate in mammals, and pigs have lower interspecies barriers compared to sparrows [21]. In 2017, Woo et al. found that PDCoV may cause respiratory infection in pigs, and in addition to fecal–oral transmission, the virus may transmit through the respiratory tract [22]. Lednicky et al. 2021 found PDCoV infection in three out of 369 plasma samples from children presenting with acute fever in Haiti [23]. Therefore, similar to many coronaviruses, PDCoV not only causes significant damage to the pig industry but also poses a potential threat to human health due to its ability for cross-species transmission and zoonotic characteristics.

On the background of SARS-CoV-2 ravaging the world, the genomic epidemiology, evolutionary dynamics, and transmission dynamics of TGEV and PDCoV, as long-emerging and new SeCoVs, respectively, can be studied to analyze the source populations, time of origin, evolutionary rate, transmission routes, and positive selection of amino acids that are essential for the S protein to help the CoVs adapt to the host. The data and results in our study will help to characterize the evolution of existing CoVs and prevent the potential risk of future cross-species transmission of CoVs to humans.

## 2. Results

### 2.1. Virus Detection and Isolation

Between 2020 and 2021, we collected 39 rectal swabs and 20 small intestinal tissue samples from piglets with diarrhea at two weeks of age from five intensive pig farms in Jiangsu and Henan provinces, China, where antibiotic treatment failed to prevent diarrheal signs in piglets. Of the 59 suspected TGEV and PDCoV positive samples in this study, three TGEV-positive samples with a positive rate of 5% and five PDCoV-positive samples with a positive rate of 8.4% were obtained. We successfully isolated a strain of TGEV and PDCoV and sequenced the complete genome with GenBank accession ON859974 and ON859973, respectively. In addition, the isolated strains were confirmed by RT-PCR (Figure 1).

### 2.2. Distribution and Phylogenetic Analysis of TGEV and PDCoV

TGEV was first reported in the United States in 1946 and has since spread to 13 countries in Europe, Asia, Africa, and South America (Figure 2a) [11]. TGEV is mainly distributed in nine provinces in eastern, northern, and central China (Figure 2b). PDCoV was first reported in Hong Kong, China, in 2012, and an outbreak in the United States also occurred in 2014, followed by a total of 11 countries worldwide (Figure 2c) [15]. PDCoV is mainly distributed in 14 provinces in eastern, northern, central, and southern China (Figure 2d).

The results of ML and BI trees based on the complete genomes of TGEV and PDCoV were generally consistent (Figure 3). TGEV was displayed as two different genotypes: genotype I and genotype II, while genotype I was divided into subtypes Ia and Ib (Figure 3a,b).

PDCoV was divided into four major lineages, namely the Southeast Asia lineage (Thailand, Vietnam, Laos), the China lineage, the USA lineage (USA, Haiti, Japan, Korea), and the early China lineage (Figure 3c,d). Among them, the Haitian isolates MW685622 and MW685624 of PDCoV were highly similar to the isolate KY065120 from Tianjin Province, China, at the nucleotide level (99.8%), and the Haitian isolate MW685623 was highly identical to the isolate KR150443 from Arkansas, USA, at the nucleotide level (98.9%).

According to the Tempest v1.5.3 regression of root-to-tip, the temporal structure analysis of TGEV after removing recombinant sequences revealed a characteristic complete genome clock-like structure (*n* = 38, correlation coefficient = 0.3485, R2 = 0.1214) (Appendix A); the linear regression of distance from root to tip against the sampling date also demonstrated that the PDCoV dataset had a significant temporal signal (*n* = 126, correlation coefficient = 0.6707, R2 = 0.4498) (Appendix A). In addition, we used BETs to evaluate the temporal signals of the TGEV and PDCoV datasets (Appendix A). The results showed that the datasets with sampling dates both outperformed the datasets with sampling dates removed, and the (log) Bayes factor (BF) calculated by the heterochronous model (Mhet) and the isochronous model (Miso) were both greater than five. The above results indicated that both the TGEV and PDCoV datasets had strong enough temporal signals to estimate the time-calibrated phylogenies.

The results of the Bayesian skyline plot (BSP) analysis demonstrated that the estimated effective population size of TGEV increased slowly from 1945 to 1960 and rapidly increased to 1000 from 1960 to 1980, corresponding to multiple outbreaks of TGEV worldwide in the last century, with slight fluctuations in the effective population size at higher levels after that time (Figure 4a).

The analysis results of the MCC trees were approximately the same as the ML and BI trees, and TGEV was divided into two genotypes: genotype I (traditional genotype) and II (variant genotype), with genotype I divided into subgenotype Ia and Ib (Figure 4b). Almost all isolates of genotype II were distributed in the USA, indicating that TGEV mainly spread in the USA and continuously underwent adaptive evolution. The appearance of PRCV could also prove this from the side. The tMRCA of TGEV was 1169.9 (948.1–1353.1, 95% highest posterior density).

From 1995 to 2011, the effective population size of PDCoV expanded at a higher rate from 1995 to 2011 and fluctuated substantially after 2011 (Figure 5a).

Due to the removal of the Southeast Asia lineage, PDCoV was displayed as two major lineages, including the USA lineage and the China lineage (Figure 5b). Reconstruction results suggested that PDCoV might have spread from the USA to China, Japan, and South Korea. The Haitian isolates also spread from China, possibly due to the hog trade between Haiti and China. PDCoV probably originated in January 1989 with a 95% highest posterior density range of March 1986–July 1991.

### 2.3. Phylogeographic Inference of TGEV and PDCoV

The worldwide spatial dispersal network of TGEV and PDCoV was reconstructed using BEAST v1.10.4 (Andrew Rambaut et al. Edinburgh, UK) (BF > 3, posterior probability >0.5). TGEV had three discrete sampling locations and four inter-country transmission routes (Figure 6a). The BSSVS results indicated that the USA was the main exporter of TGEV and the importers were China (BF = 11.162, migration rate = 0.6305) and Mexico (BF = 11050.891, migration rate = 0.6785) (Figure 6c), which was consistent with the analysis of the MCC tree. Meanwhile, China (BF = 3.667, migration rate = 2.354) and Mexico (BF = 3.243, migration rate = 0.7824) have imported TGEV to the USA (Appendix A). The USA and Mexico share a border close to 3169 km in length with close communication. The USA is a major exporter of hog, so the transmission route from the USA to Mexico has a high BF value and migration rate (Figure 6b,d). These two sampling locations have a strong signal of mutual transmission. China and the USA are 11,172 km apart, and although the communication distance is relatively far, both countries are the largest trading countries, so the migration rate from China to the USA and the BF value from the USA to China are both high. Therefore, the emergence of variant TGEV strains in the USA may be caused by the recombination of TGEV in the USA due to the USA being both an exporting and an importing country, and the emergence of PRCV and the production of TGEV and PEDV chimeric viruses may confirm this. Therefore, combining the results of the MCC tree and the BSSVS analysis, it is likely that TGEV originated in North America, most likely in the USA, and thus radiated to Asia, South America, Europe, and Africa.

For PDCoV, there were six discrete sampling locations and eight transmission routes according to BSSVS results (Figure 7a). The USA was the dominant exporting country and the spread was to four locations, namely China (BF = 31.989, migration rate = 1.318), Haiti (BF = 31.851, migration rate = 0.94), Japan (BF =28494.598, migration rate = 2.494) and South Korea (BF =195.023, migration rate = 1.753) (Figure 7c). China was associated with two locations, Haiti (BF = 31.036, migration rate = 0.487) and the USA (BF = 95.023, migration rate = 0.53). Moreover, there were two routes from Japan to South Korea (BF = 10.865, migration rate = 1.146) and South Korea to Peru (BF = 8.094, migration rate = 1.049) (Appendix A). Japan and Korea are adjacent to each other at a distance of 1151 km, so this route has a high BF value and high migration rate (Figure 7b,d). The PDCoV input country for Japan and Korea was probably the USA, consistent with the results of the MCC tree. The distance between Japan and the USA is 10,162 km with a high BF value and high migration rate, indicating a strong signal in this route, confirming that the source of PDCoV in Japan was the USA. The sources of PDCoV for Haiti were probably the USA and China. Due to the shorter communication distance and closer hog trade, the BF value and migration rate from the USA to Haiti were higher, so the USA was more likely to be the PDCoV input country for Haiti. Combining the results of the MCC tree and the BSSVS analysis, it is likely that PDCoV originated in Asia in 1989, most likely in China, and subsequently spread to the United States and then radiated to other countries.

### 2.4. Adaptive Evolution Sites and Structural Analysis of the S Protein

Based on the results displayed by Datamonkey, we identified one positive selection site in the TGEV S protein (site 218) and seven positive selection sites in the PDCoV S protein (sites 110, 123, 137, 527, 630, 642, and 1016) (Appendix A). The positive selection sites of the TGEV S protein were not visualized because there was no available PDB file for the TGEV S protein crystal structure. Currently, it is known that site 218 is not in the receptor-binding domain (RBD, aa 506–655), and the zoonotic potential of CoVs is determined by the receptor-binding properties of the S protein, making it unlikely that cross-species transmission of TGEV will occur. Deletion of the structural domain 0 (aa 145–155) responsible for attachment of sialoglycans from the TGEV S gene also results in the production of PRCV, which, in turn, leads to loss of enteric tropism [24].

The identified sites of the PDCoV S protein were visualized in UCSF Chimera X (Figure 8). The S protein is a homotrimer consisting of S1 and S2 subunits (Figure 8b). The NTD of the S1 subunit (shown in red) contains the RBD (aa 300–419), and the S2 subunit (shown in green) is responsible for the fusion of the virus with the cell membrane. Four sites under selection were located in the S1 subunit and three in the S2 subunit. Amino acid sites 110, 123, and 137 in the S1 subunit were located in the NTD, and mutation of site 137 might eliminate specific van der Waals forces, thereby enhancing the flexibility of the S1 subunit and possibly accelerating membrane fusion events for viral transmission. The main chain atom of the amino acid Tyr 123, located in the α-helix, formed a hydrogen bond with Ala 119 (3.037 Å); the main chain atom of the amino acid Ala 137, located in the loop between two β-folds, formed a hydrogen bond with Thr 136 (2.619 Å), which might stabilize the loop.

The S2 subunit contains the conserved protease cleavage sites Arg 673 and Arg 669; consistent with PDCoV, it requires trypsin cleavage to allow viral transmission and PDCoV is exposed to high concentrations of proteases in the intestine of infected pigs [25]. The amino acid Ala 630 under selection was located in the central helix N (CH-N), which was close to Leu 720 located in the fusion peptide (6.295 Å) and formed hydrogen bonds with Leu 626 (2.955 Å) and Ser 634 (2.938 Å), which were also in the α-helix, and Gln 592 (3.360 Å), which was in the loop.

## 3. Discussion

Multiple outbreaks of zoonotic coronaviruses in this century have raised worldwide concern about coronaviruses, especially the COVID-19 outbreak that raised alarm regarding cross-species transmission of coronaviruses (https://covid19.who.int/ (accessed on 5 April 2022)) [26,27]. The report of PDCoV infection in humans in 2021 and the emergence of the world’s first SARS-CoV-2 transmission to humans via cats indicate that coronaviruses significantly threaten human public health [6]. Since TGEV can alter tissue tropism and form chimeric viruses with other coronaviruses [13,14], studies on the epidemiology and transmission mechanisms of SeCoV provide lessons for the prevention and control of other coronaviruses. Finally, we selected the once extremely popular TGEV and the current emerging PDCoV as the subjects and studied the evolution and transmission characteristics.

We collected 59 samples from Jiangsu and Henan provinces in China, successfully isolated a strain of TGEV and PDCoV, and sequenced the complete genome. Based on the phylogenetic results, TGEV was classified into genotypes I and II. Genotype II was divided into subtypes Ia and Ib; PDCoV was displayed as four lineages: Southeast Asia, USA, China, and early China. The presence of TGEV genotype II (variant) almost exclusively in the USA suggested that TGEV had undergone adaptive evolution in the USA [11], as evidenced by the presence of PRCV and chimeric TGEV and PEDV strains. The similarity of the Haitian PDCoV strain to the USA lineage strain might be since Haiti relied heavily on the hog trade after the African swine fever outbreak. Thus the USA was likely the exporting country of PDCoV to Haiti [23,28].

Combining the results of the MCC trees and BSSVS in this study, the TGEV might have originated in 1169, and the PDCoV may have originated in 1989. In addition, the Bayesian skyline plot analysis results indicated that the effective population size of TGEV experienced rapid growth between 1960 and 1970 and fluctuated slightly at higher levels after 1980. The effective population size of PDCoV increased rapidly until 2011 and then fluctuated briefly and stabilized. In addition, we analyzed the linear regression relationship between the geographic distance and migration rate. We found that the distance between China and the USA showed a strong signal of virus transmission (migration rate = 2.354, distance = 11,172 km). The distance between the USA and Japan (migration rate = 2.494) and South Korea (migration rate = 1.753) was about 11,000 km, but there was also a strong signal of virus transmission. This suggested that there was no direct association between geographic distance and the migration rate of virus transmission, and the linear regression relationship between distance and migration rate does not show a certain trend (Appendix A). The above phenomenon might be due to frequent hog trade between countries or contamination caused by various factors that alter the natural rate of virus transmission [29,30].

As the most important structural protein of CoVs, changes in the structure and function of the S protein affect the histophilicity and membrane fusion of the viruses [31,32,33], which are more likely for cross-species transmission. We thus analyzed the effect of positive selection on TGEV and PDCoV S proteins. We found that the positive selection site of the TGEV S protein was amino acid 218, and since it was not in the RBD, cross-species transmission of TGEV was unlikely to occur. In addition, genotype I might be more suitable for transmission between pigs than genotype I [34]. Seven positive selection sites of the PDCoV S protein were identified, and mutations in these amino acids might enhance the S protein’s flexibility and membrane fusion activity, thereby accelerating virus transmission.

There are many limitations to this study. Firstly, due to the sampling bias in the datasets, this affects the Bayesian phylogenic and inference [35]. Secondly, the dataset of TGEV only has sequences from three sites, the USA, China, and Mexico. At the same time, it is a fact that many countries, including Europe and Africa, have TGEV outbreaks but many regions do not upload the corresponding sequences in GenBank. The dataset of PDCoV also has the same problems. Thirdly, there is no available S protein PDB file for TGEV, so it was not demonstrated in this study, as there are few studies targeting the TGEV protein structure. Finally, the use of only 59 samples to generalize to the entire swine or human population is limited, especially since these viruses are widely spread in both species and in other animals. The results obtained in this study do not serve as predictions for future SeCoV outbreaks; rather we can forecast using scientific evidence from the past.

In conclusion, our work provides new insights into the transmission characteristics and origins of TGEV and PDCoV. It is the first study on the evolutionary dynamics and phylogeography of TGEV. An increasing number of studies have demonstrated the ability of CoVs to spread across species and become zoonotic viruses [36,37,38]. As an economically important animal for humans, pigs may serve as a container for SeCoVs and other CoVs and thus spread them to other organisms [21,39]. Therefore, increased surveillance and research on pigs are needed to prevent future pandemics of CoVs.

## 4. Materials and Methods

### 4.1. Sample Collection and Virus Isolation

Between 2020 and 2021, 39 rectal swabs and 20 small intestine samples were collected from five intensive pig farms in two cities in northern Jiangsu Province and three cities in northeastern Henan Province, China. Virus isolation was carried out in swine testis (ST) cells (ATCC CRL1746) and LLC porcine kidney (LLC-PK) cells (ATCC CL-101). The above cells grew in Dulbecco’s modified eagle medium (DMEM, Gibco, Waltham, MA, USA) supplemented with 8% heat-inactivated fetal bovine serum (FBS, Gibco) and 1% penicillin–streptomycin (Gibco, Waltham, MA, USA). For the isolation of TGEV and PDCoV, the trypsin (Gibco, Waltham, MA, USA) concentration in DMEM was adjusted to 1 μg/mL and 10 μg/mL, respectively.

We confirmed the successful isolation of the virus by cytopathogenic effect (CPE) and reverse transcription-polymerase chain reaction (RT-PCR; Appendix A).

### 4.2. Viral RNA Extraction and Genome Sequencing

According to the instruction manual, viral RNA was extracted using the E.Z.N.A Viral RNA Kit (OMEGA Bio-Tek). The cDNA was synthesized using Reverse Transcriptase M-MLV (Takara, Dalian, China) and Random Primer (Takara, Dalian, China). We designed 12 and 10 pairs of specific primers for full-length genomic amplification of TGEV and PDCoV positive samples, respectively (Appendix A), by Platinum SuperFi II high-fidelity DNA polymerase (Invitrogen, Waltham, MA, USA). PCR amplification products were ligated using DNA A-Tailing Kit (Takara, Dalian, China) and pMD18-T vectors (Takara, Dalian, China), and finally, all plasmids were sent to Tsingke Biotechnology Co., Ltd. (Beijing, China). for Sanger sequencing.

### 4.3. Sequence Analysis and Phylogenetic Analysis

We retrieved all available complete genome sequences of TGEV (txid: 11149, slen: 27000–29000) and PDCoV (txid: 1586324, slen: 25000–26000) from the National Center for Biotechnology Information (NCBI) by taxonomy ID and nucleotide length as of 5 April 2022. (Appendix A). The downloaded sequences were filtered and organized by Phylosuite v1.2.2 (Dong Zhang et al. China) [40]. All sequences were input to MAFFT v7.313 (Research Institute for Microbial Diseases, Osaka University, Suita, Japan) [41] for alignment, then aligned sequences were loaded to Gblocks v0.91b (Jose Castresana et al. Spain) [42] to determine conservative domains and adjust them manually. ModelFinder v1.6.8 (Lars S Jermiin et al. Vienna, Austria) [43] was used to test and select the most suitable nucleotide substitution models. Different model combinations were set up by combining molecular clock models (Strict and UNCL) and tree prior (Coalescent: Constant Size and Bayesian Skyline) [44]. The values of marginal likelihood estimation for all combinations were obtained by the path sampling/stepping stone sampling method, and the combination with the most significant value was selected as the best.

Maximum likelihood (ML) trees based on TGEV and PDCoV complete genome sequences were constructed according to the corrected Akaike information criterion by IQ-TREE v1.6.8 (Lars S Jermiin et al. Vienna, Austria) [45] with 1000 bootstraps, where the general time-reversible substitution models were GTR+F+R2 (TGEV) and GTR+F+R4 (PDCoV). Bayesian inference (BI) trees based on TGEV and PDCoV complete genome sequences were constructed according to the corrected Akaike information criterion by Mrbayes v3.2.6 (F Ronquist et al. Rochester, USA) [46] with 2,000,000 generations, modeled as the best fit general time-reversible substitution model with a proportion of invariant sites, empirical base frequencies, and gamma-distributed rate heterogeneity (GTR+F+I+G4).

Identification of recombination events in complete genome sequences of TGEV and PDCoV by all seven methods in RDP4 v 4.101 [47] (RDP, GENECONV, 3Seq, Chimaera, SiScan, MaxChi, and BoostScan) and detection of recombination by at least three methods with a *p*-value cutoff of 0.05 was considered as true recombination, removing recombination sequences until recombination events were no longer detected. Initial inspection of the temporal symbols in TGEV and PDCoV datasets was performed by Tempest v1.5.3 [48]. Finally, it was examined by Bayesian evaluation of temporal signal (BETs) to remove the invalid clades. The most recent common ancestor (tMRCA) and evolutionary rates were estimated using the BEAST package v1.10.4 (Andrew Rambaut et al. Edinburgh, UK) [49], with a GTR+G4 nucleotide substitution model, a UNCL clock model, and the Coalescent Bayesian Skyline tree prior. Markov chain Monte Carlo (MCMC) chains were run for the TGEV dataset with 200,000,000 steps, sampled every 20,000 steps; the PDCoV dataset was obtained by combining four independent datasets with 500,000,000 steps, sampled every 50,000 steps, by LogCombiner v1.10.4 (Andrew Rambaut et al. Edinburgh, UK), with a final sample size of over 10,000 to ensure that all parameters converged and the effective sample size (ESS) > 200, burning up the first 10% of total chain length. Maximum clade credibility (MCC) trees were generated by TreeAnnotator v1.10.4 (http://tree.bio.ed.ac.uk/software/figtree/ (accessed on 5 April 2022)), and Bayesian skyline plots were constructed in Tracer v1.7.2 (http://beast.community/tracer (accessed on 6 April 2022)) to estimate population dynamics of TGEV and PDCoV [50]. Lastly, all final trees were edited and illustrated in Figtree v1.4.4 (http://tree.bio.ed.ac.uk/software/figtree/ (accessed on 8 April 2022)).

### 4.4. Phylogeographic Analyses of TGEV and PDCoV

The pre-processed dataset from 2.3 was imported into BEAST v1.10.4, and the BEAGLE library [51] was used to enhance computational performance for phylogeography analysis. Based on the results of path sampling and stepping stone sampling (Appendix A), we used a combination of the GTR+G nucleotide substitution model, UNCL molecular clock model, and Bayesian stochastic search variable selection (BSSVS) traits model. The chain length and sampling parameters were the same as in 2.3, with a final sample size of over 10,000 to ensure that all parameters converged and the effective sample size (ESS) > 200, burning up the first 10% of total chain length. The trees were generated by TreeAnnotator and illustrated in Figtree v1.4.4. The phylogeography analysis results were generated by Spatial Phylogenetic Reconstruction of Evolutionary Dynamics 3 (SPREAD3) v0.9.7 (Philippe Lemey, Leuven, Belgium) [52].

### 4.5. S protein Positive Selection and Functional Analysis

Datasets of TGEV and PDCoV without recombinant sequences were imported into Datamonkey (http://www.datamonkey.org/ (accessed on 4 May 2022)) for analysis of the positive selection sites of the S gene. The methods used in Datamonkey to infer positive amino acid sites included single likelihood ancestor counting (SLAC), fixed effects likelihood (FEL) [53], mixed effects model of evolutionary (MEME) [54], and fast unconstrained Bayesian approximation (FUBAR) [55]. Codons were considered under selection if at least three methods identified them. There was no available S protein PDB file for TGEV, and we selected PDB file 6B7N to demonstrate the S protein crystal structure of PDCoV via UCSF Chimera X (Thomas E Ferrin et al. San Francisco, USA) [56].

## Figures and Tables

**Figure 1 ijms-23-09786-f001:**
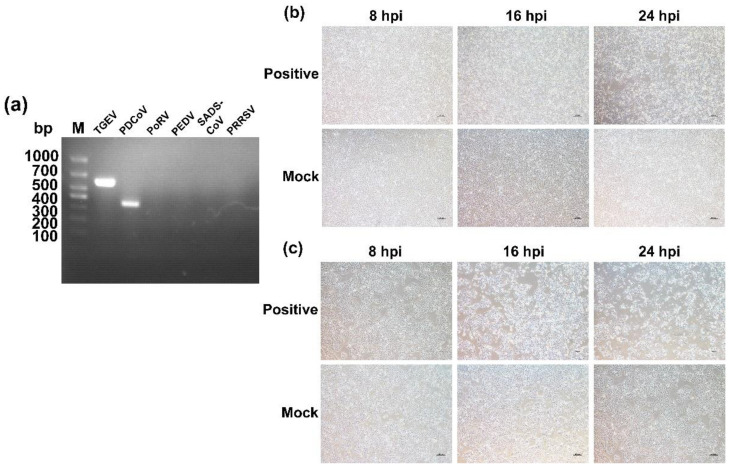
Isolation and identification of TGEV and PDCoV. (**a**). RT-PCR was performed on the samples to ensure no other common viral contamination. (**b**). Isolation and passage of TGEV in ST cells (40×). (**c**). Isolation and passage of PDCoV in LLC-PK cells (40×).

**Figure 2 ijms-23-09786-f002:**
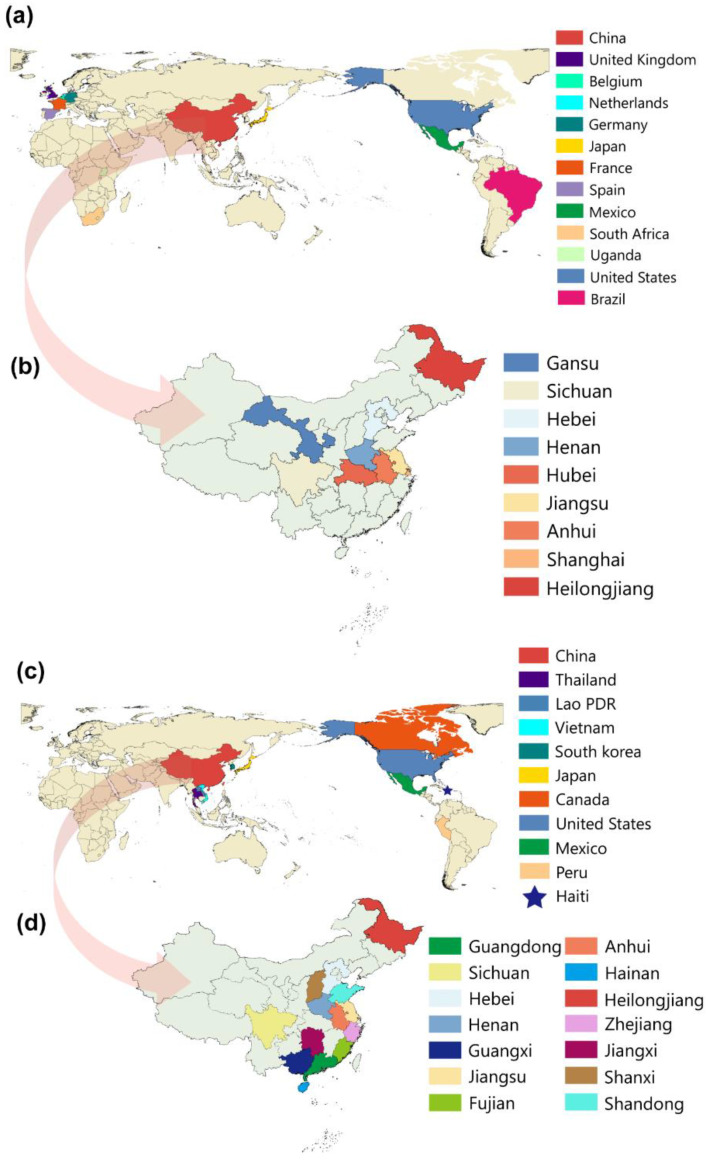
Geographical distribution of TGEV and PDCoV around the world and in China. (**a**). Different colors characterize the distribution of TGEV in different countries. (**b**). Distribution of TGEV in China, different provinces were indicated by different colors. (**c**). Different colors characterize the distribution of PDCoV in different countries. (**d**). Distribution of PDCoV in China, different provinces were indicated by different colors.

**Figure 3 ijms-23-09786-f003:**
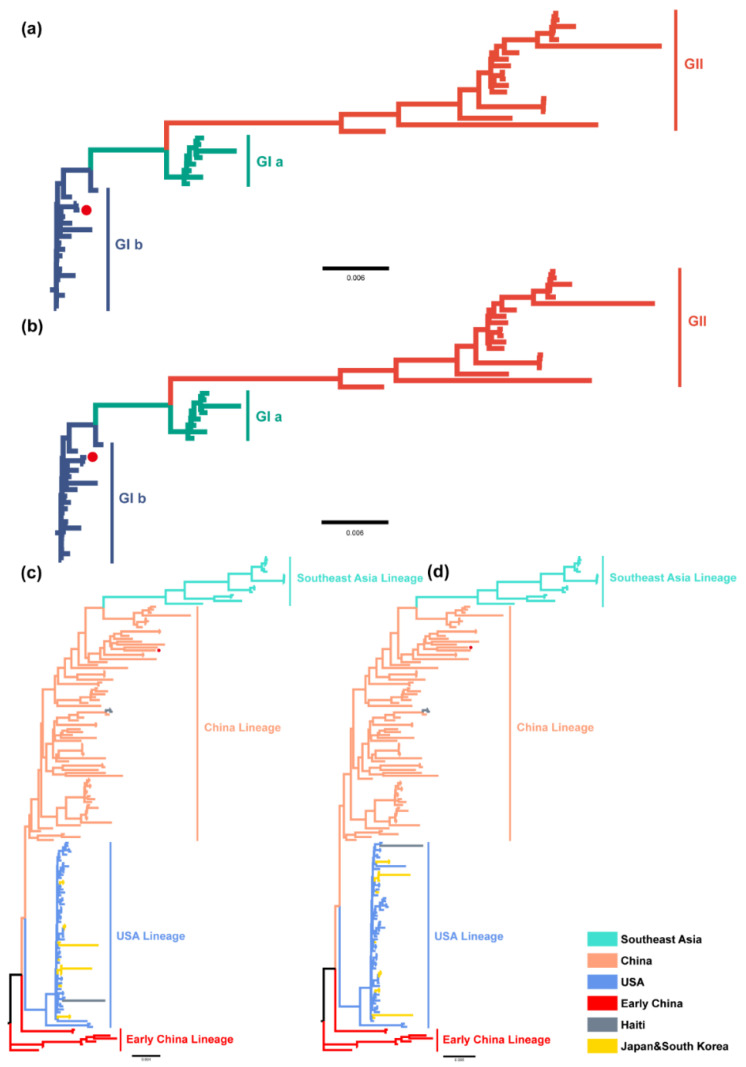
ML tree and BI tree of TGEV and PDCoV. (**a**). ML tree of TGEV complete genome. (**b**). BI tree of TGEV complete genome. Different colors represented different genotypes. (**c**). ML tree of PDCoV complete genome. (**d**). BI tree of PDCoV complete genome. Different colors indicated different genotypes/lineages.

**Figure 4 ijms-23-09786-f004:**
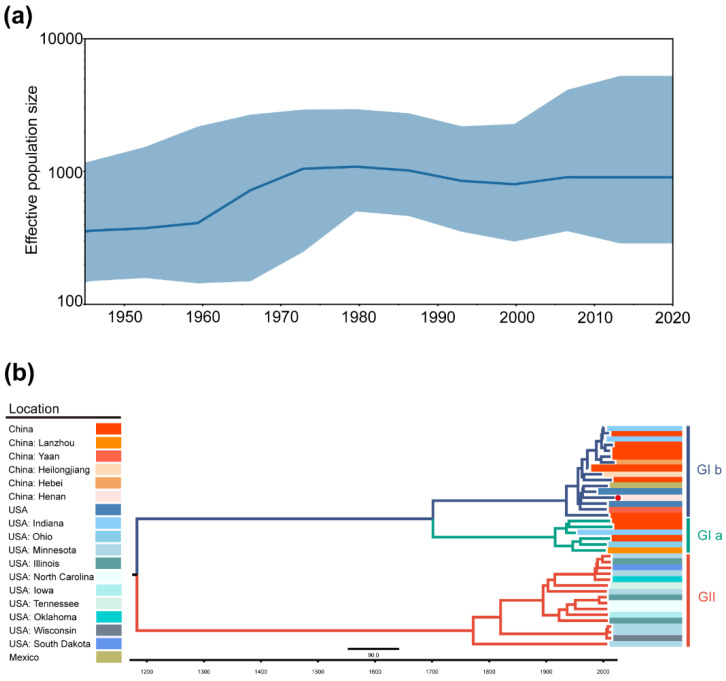
Demographic history of TGEV. (**a**). Demographic history was inferred via Bayesian skyline analysis. The median and 95% HPD intervals were plotted. (**b**). MCC tree of TGEV complete genome. Different colors represented different genotypes. The red dot indicates that the strain was sequenced in this study.

**Figure 5 ijms-23-09786-f005:**
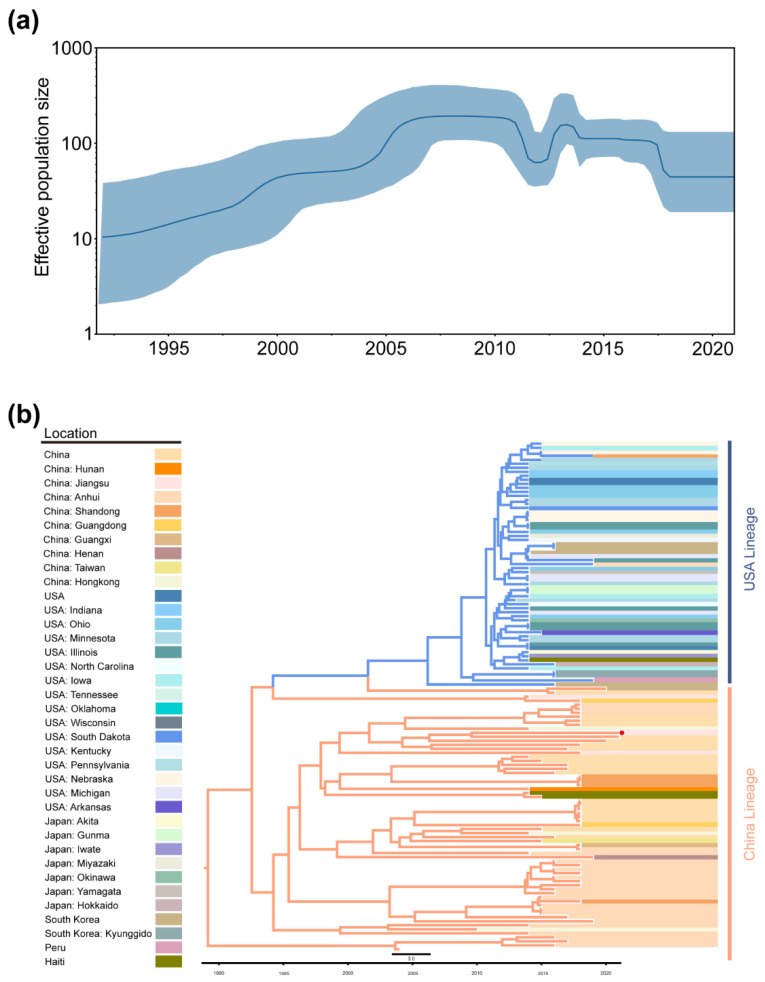
Demographic history of PDCoV. (**a**). Demographic history was inferred via Bayesian skyline analysis. The median and 95% HPD intervals were plotted. (**b**). MCC tree of PDCoV complete genome. Different colors represented different lineages (we merged early China lineage into China lineage). The red dot indicates that the strain was sequenced in this study.

**Figure 6 ijms-23-09786-f006:**
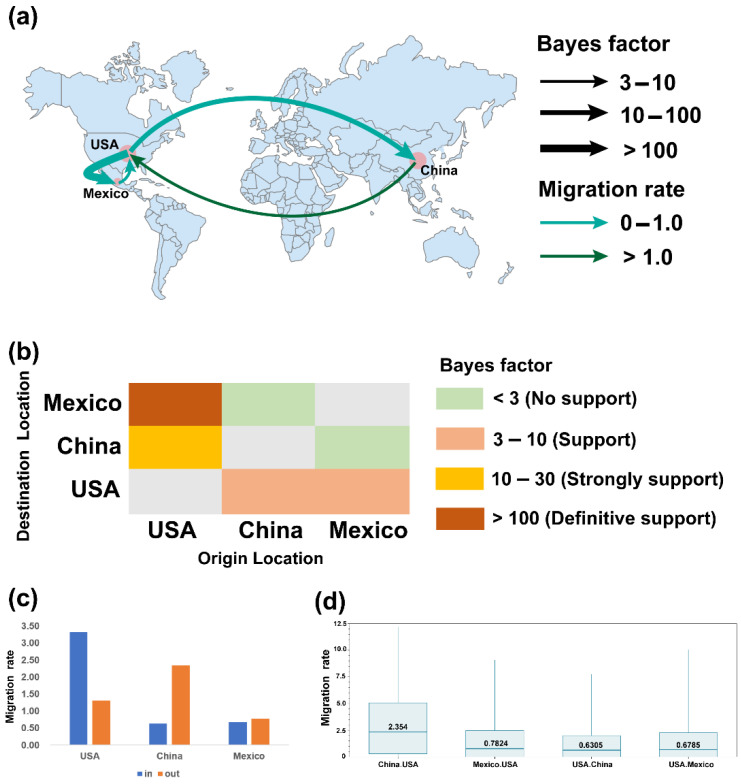
BSSVS analysis of TGEV. (**a**). Phylogeographic reconstruction of estimated global spatial diffusion of TGEV. The curves and arrows indicate the direction and geographical location of TGEV migration (BF > 3, posterior probability > 0.5). The color and width of the curves represent the BF value and migration rate, respectively. (**b**). BF supports between the USA, China, and Mexico. (**c**). The histogram of TGEV migration changes for each location. (**d**). Migration rates for each supported migration route.

**Figure 7 ijms-23-09786-f007:**
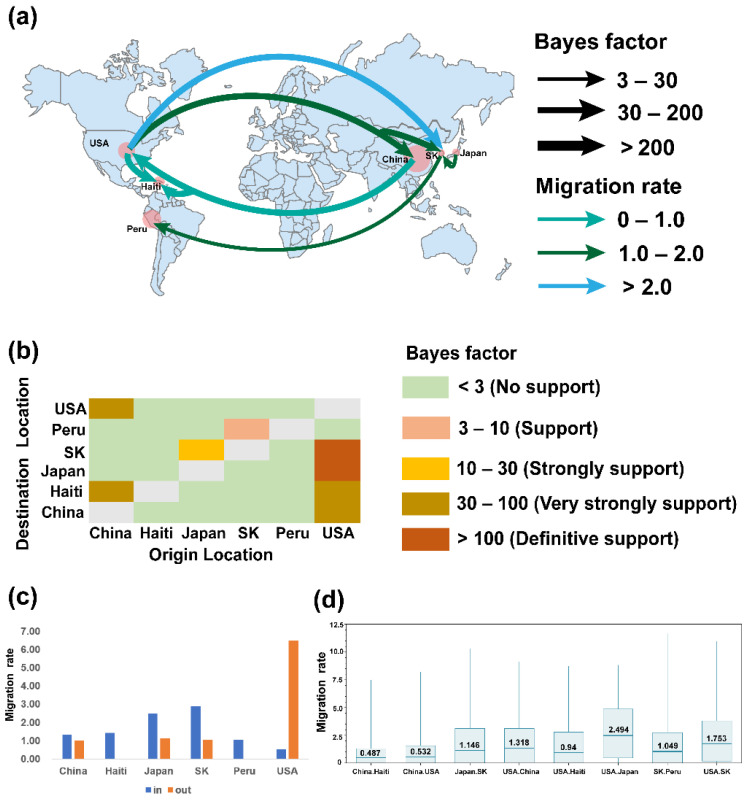
BSSVS analysis of PDCoV. (**a**). Phylogeographic reconstruction of estimated global spatial diffusion of PDCoV. The curves and arrows indicate the direction and geographical location of PDCoV migration (BF > 3, posterior probability > 0.5). The color and width of the curves represent the BF value and migration rate, respectively. (**b**). BF supports between China, Haiti, Japan, SK, Peru, and the USA. (**c**). The histogram of PDCoV migration changes for each location. (**d**). Migration rates for each supported migration route. South Korea, SK.

**Figure 8 ijms-23-09786-f008:**
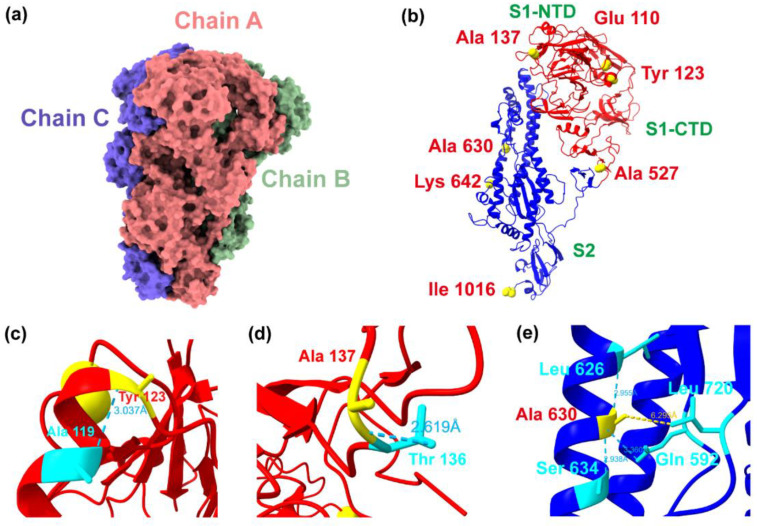
Structural display of the PDCoV S protein and the location of positive selection sites in the structure. (**a**). Surface representation of PDCoV S-trimer. (**b**). Cartoon representation of PDCoV S monomer. The S1 subunit is shown in red and S2 subunit in blue. Selected amino acids are presented as yellow spheres. (**c**). Hydrogen bond exists between Tyr 123 and nearby Ala 119 (shown in cyan). (**d**). Hydrogen bond exists between Ala 137 and Thr 136 (shown in cyan). (**e**). Ala 630 formed hydrogen bonds with Leu 626, Ser 634, and Gln 592, close to Leu 720 located in the fusion peptide (6.295 Å).

## Data Availability

NCBI Bioproject: PRJNA852690 and PRJNA851852.

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
