# Peer review of "Historical Evolutionary Dynamics and Phylogeography Analysis of Transmissible Gastroenteritis Virus and Porcine Deltacoronavirus: Findings from 59 Suspected Swine Viral Samples from China"

_ijms, 2022, doi:10.3390/ijms23179786_

Round 1

Reviewer 1 Report

The aim of this submission is to present a conducted study to characterize the evolution of existing CoVs and prevent the potential risk of future cross-species transmission of CoVs to humans.  The study and its findings can have a significant contribution to the current attempt to understand the ecology of the coronaviruses as a family.  Below are my general and specific comments with their related suggestions:

1.        The title of the submission is much broader than its contents as well as the aim of the study.  The study was limited in its scope to the available isolates by production type and geographical location.  Therefore, the inference from such study is limited.  I may suggest you delete the” past and future of swine enteric coronavirus” from the title  or replace it with “findings from 59 suspected swine viral samples from China”. If you add the suggested words, then there is a need to insert “historical” prior to evolutionary to cover the other historical section of this study

2.       Some of the presented results referred to the samples as suspected isolates, others as positive.  Please be consistent and define the clinical meaning of suspected samples.  Are these samples , for example, from healthy or clinical sick pigs?

3.       The submission requires further details of the study design particularly in describing the sources of the farms for the 59 isolates with geographical locations.  Literature searching methods for the sources of the historical data should be also mentioned under the study design/materials and methods.

4.       The authors used the term ’Symptoms” in referring to clinical signs in pigs.  Animals do not show symptoms to humans.  Please replace symptoms with clinical signs.

5.       The used term “future” of the evolution findings is inappropriate.  Scientists or others cannot predict the future; rather we can forecast using scientific evidence from the past.  I therefore suggest inserting 1-2 sentences to indicate so under the limitations of this study.

6.       The use of only 59 viral isolates is limited to generalize to the entire swine or human populations particularly with the wide spread of these viruses across these two species among other mammals.  Therefore, this limitation should be stated under the discussion session.

In conclusion this submission requires major revision by considering the above comments with a second review prior to a final decision for publication. 

Author Response

Dear reviewer:
Thank you for your valuable comments and suggestions. Based on your comments and suggestions, we make the following changes:
response:

Line 1 - The title "wraped" was changed to "Historical evolutionary dynamics and phylogeography analysis of transmissible gastroenteritis virus and porcine delta coronavirus: findings from 59 suspected swine viral samples from China";

Line 77 - The word "Symptoms" was changged to "signs"; 

Line 105 - The sentence "Between 2020 and 2021, we collected 39 rectal swabs and 20 small intestinal tissues from piglets with diarrhea at two weeks of age in 5 intensive pig farms in Jiangsu and Henan provinces, China, where antibiotic treatment failed to prevent diarrheal signs in piglets." was added.

Line 356-362 The sentences added in discussion are the responses to the fifth and sixth comments. 

Line 372 The first sentence is the response to the third comment. 

Line 392-395 We added the details of the literature searching methods for the sources of the historical data.

The above changes are all rendered in Word in revision mode.

Thank you again for your responsible and detailed suggestions.

Reviewer 2 Report

In this manuscript, authors analyzed genome sequences of TGEV and PDCoV by evolutionary dynamics and phylogeography, revealing the genetic diversity and Spatio-temporal distribution characteristics. The results provide new insights into the evolutionary characteristics, transmission diversity, and the potential for cross-species transmission of TGEV and PDCoV. The findings are of great reference value and are recommended to be published directly in IJMS.

Author Response

尊敬的审稿人:
感谢您提出宝贵的意见和建议。 

Round 2

Reviewer 1 Report

Thanks for considering the suggestions by the reviewers.  Please review the revision with professional English editor prior to its finalization.  Congratulations  for the acceptance of this submission to be published.  

Sincerely 

Mo Salman, Editor